# Cemiplimab in Ultra-Octogenarian Patients with Cutaneous Squamous Cell Carcinoma: The Real-Life Experience of a Tertiary Referral Center

**DOI:** 10.3390/vaccines11091500

**Published:** 2023-09-18

**Authors:** Nerina Denaro, Emanuela Passoni, Alice Indini, Gianluca Nazzaro, Giada Anna Beltramini, Valentina Benzecry, Giuseppe Colombo, Carolina Cauchi, Cinzia Solinas, Mario Scartozzi, Angelo Valerio Marzano, Ornella Garrone

**Affiliations:** 1Oncology Unit, Fondazione IRCCS Ca’ Granda, Ospedale Maggiore Policlinico, 20122 Milan, Italy; carolina.cauchi@policlinico.mi.it (C.C.); ornella.garrone@policlinico.mi.it (O.G.); 2Dermatology Unit, Fondazione IRCCS Ca’ Granda, Ospedale Maggiore Policlinico, 20122 Milan, Italy; emanuela.passoni@policlinico.mi.it (E.P.); gianluca.nazzaro@policlinico.mi.it (G.N.); valentina.benzecry@policlinico.mi.it (V.B.); angelovalerio.mazzano@policlinico.mi.it (A.V.M.); 3Melanoma Unit, Department of Medical Oncology and Hematology, Fondazione Istituto di Ricovero e Cura a Carattere Scientifico, Istituto Nazionale dei Tumori, 20133 Milan, Italy; alice.indini@istitutotumori.mi.it; 4Maxillofacial Surgery and Odontostomatology Unit, Department of Biomedical Surgical and Odontoiathric Science, Fondazione IRCCS Ca’ Granda, Ospedale Maggiore Policlinico di Milano, 20122 Milan, Italy; giada.beltramini@policlinico.mi.it (G.A.B.); giuseppe.colombo@policlinico.mi.it (G.C.); 5Department of Biomedical, Surgical and Dental Sciences, Università degli Studi di Milano, 20122 Milan, Italy; 6Medical Oncology Department, University of Cagliari, 09126 Cagliari, Italy; czsolinas@gmail.com (C.S.); marioscartozzi@gmail.com (M.S.); 7Department of Pathophysiology and Transplantation, Università degli Studi di Milano, 20122 Milan, Italy

**Keywords:** ultra-octogenarian, cutaneous squamous cell carcinoma, Cemiplimab, immunotherapy, geriatric oncology

## Abstract

Background: The incidence of cutaneous squamous cell carcinoma (cSCC) is rapidly increasing, paralleling the aging of the population. cSCC predominantly affects chronically sun-exposed areas, such as the head and neck region. At our tertiary center, a multidisciplinary approach to non-melanoma skin cancer is provided for locally advanced cSCC. Methods: We retrospectively revised all patients with locally advanced/metastatic cSCC treated with anti-PD1 antibody (Cemiplimab) at our Institution from January 2020 to March 2023 (minimum follow-up of 4 months on treatment). Results: Overall, we consecutively treated 20 ultra-octogenarian patients, of whom 15 were males and 5 were females (median age: 86.9 years). Despite age, a median number of concomitant drugs, and comorbidities, efficacy, and safety were superimposable with the available literature. No patients reported treatment-related adverse events of grade 3 or higher. Grade 2 adverse events were reported in 25% of patients. Overall, the response rate was 65%, with 50% partial responses and 20% long-lasting stable disease. The median duration of response was 14 months. The G8 elderly score was assessed in all patients, and the median score was 12 (range 9–14). Conclusions: Among ultra-octogenarian patients, a clinical benefit from Cemiplimab was obtained in most, including tumor shrinkage and pain relief. Cemiplimab confirmed its effectiveness in elderly patients in a real-life setting, with no new safety concerns.

## 1. Introduction

Cutaneous squamous cell carcinoma (cSCC) is the second most common skin cancer, and its incidence is rapidly increasing in parallel with the aging of the population [1].

Common risk factors for cSCC are older age, immunosuppression, and chronic sun exposure. Surgery is the mainstay for early-stage (i.e., localized) disease, while locally advanced disease requires a multidisciplinary assessment [2].

Immunosuppression is strongly associated with an increased risk of developing cSCC. Transplant recipients have a 65–250-fold increase in risk of cSCC, and a 10-fold increase in risk. [3,4,5] The overall median DOR was 41.3 months. Fatigue (any grade) was the most common treatment-emergent adverse event (TEAE) (34.7%), and hypertension (4.7%) was the most common Grade ≥3 TEAE. The median progression-free survival (PFS) was 22.1 months. However, ENPOWER CSCC1 did not include a special population of patients. The exclusion criteria included the following: autoimmune disease requiring systemic immunosuppressant agents within 5 years; history of solid organ transplant; history of pneumonitis within the last 5 years; active infection requiring therapy, including known infection with human immunodeficiency virus, or active infection with hepatitis B or hepatitis C virus; chronic lymphocytic leukemia (CLL); brain metastases; and the Eastern Cooperative Oncology Group (ECOG) performance score (PS) ≥ 2 [6,7].

Despite there being no clear definition for “special population”, some clinical situations are a challenge, especially among elderly patients. Nevertheless, real-life experience and data on frail or elderly patients and special populations have suggested they might be candidates for cancer immunotherapy, but their management requires a multidisciplinary approach and a comprehensive geriatric assessment (CGA) [3,4,5,6,7,8]. CGA is a multiparametric evaluation considering functional capacity, cognitive function and mood, polypharmacy, and social and financial support [8]. CGA is not available in all settings due to issues related to the time required for evaluation and the need for the coordination of multidisciplinary specialties. The two major tools to identify patients at higher risk of developing functional decline and disability are the Geriatric 8 (G8) and FRAIL scales (Fatigue, Resistance, Ambulation, Illnesses, Loss of weight) (Table 1) [9,10]. The identification of comorbidities and kidney/cardiovascular or neurologic impairment that may increase the risk of anticancer treatment toxicities is essential to better assess the risk-benefit ratio of anti-cancer treatments [8]. To the best of our knowledge, no specific validation of the G8 and FRAIL scales in ultra-octogenarian is published. Indeed, in our clinical practice, only a geriatric assessment including the G8 and FRAIL scales and multidisciplinary clinical evaluation has been offered to all patients [9,10].

In this paper, we revised real-world experience on Cemiplimab outcomes in elderly patients with locally advanced/metastatic LAcSCC/mcSCC treated at our referral center. We also report our experience in building a multidisciplinary team (MDT) for the management of patients with skin cancer, and related challenges.

## 2. Methods

We conducted a retrospective observational study to assess real-world treatment compliance and outcomes in ultra-octogenarian patients with unresectable LA-cSCC or mcSCC. We used the electronic health records of patients who were treated with Cemiplimab from January 2020 to May 2023.

The inclusion criteria were the following: age ≥ 80 years at the time of systemic treatment initiation; systemic treatment with Cemiplimab (at least 2 doses); and informed consent to data collection processing and privacy. The exclusion criteria were the following: age < 80 years; major contraindications to antiPD1 therapy; and patients’ refusal of Cemiplimab treatment.

At our Institution, patients with locally advanced, recurrent, or metastatic cSCC are all evaluated by a skin cancer multidisciplinary team (MDT). The members of the skin MDT are dermatologists, radiation oncologists, medical oncologists, maxillo-facial surgeons, plastic surgeons, and pathologists. On-demand geriatricians and nutritionists are likewise involved.

The Geriatric 8 (G8) questionnaire was administered prior to the start of treatment [9]. We perform the FRAIL scale test to correlate the G8 score with the Frailty assessment tool, prior to the start of treatment. In the FRAIL scale, the number of illnesses is considered Illness * (0–4 = 0; 5–11 = 1).

Adverse events (AEs) intensity was graded according to the Common Terminology Criteria for Adverse Events (CTCAE) version 5.0.

Data collection was performed under normative regulations, indications, and restrictions on the matter of retrospective clinical studies.

All patients provided written informed consent according to the actual rules for data collecting, processing, and privacy required in Italy.

## 3. Results

Overall, 20 patients who fulfilled the inclusion criteria were consecutively treated and enrolled in the present study. Table 2 and Table 3 summarize the patients’ characteristics and treatments.

The main patient characteristics were the following: 15/5 M/F; a median age at treatment start of 86.9 years (range 80–103); and a median number of surgeries of 2 (range 0–4). Fourteen patients (79%) received prior radiotherapy for the primary cSCC after first diagnosis or at the time of disease recurrence.

Overall, the G8 median score was 12 (range 9–14). Eight (40%) patients were assessed as vulnerable by the G8 score. These patients were evaluated by the MDT and referred for CGA. However, 6 out of 8 patients with G8 scores ≤ 14 did not receive a CGA because at that time, an onco-geriatric pathway was not available at our Institution.

All six vulnerable patients who did not undergo CGA received at least two cycles of Cemiplimab. The median number of cycles at the time of the present analysis was 14.

We tried to correlate the G8 score and the FRAIL score, however while the first allowed us to personalize the geriatric risk (with a median value of 12, range 1–17), the FRAIL score with a small range of 1–5 and a median value of 2 was used to indicate a yes or no condition.

Seven patients (35%) suffered from metabolic syndrome, eleven patients (55%) had cardiovascular disease history, and five patients (25%) had chronic pulmonary disease. Two patients received Cemiplimab while on immunosuppressant therapy for hematologic disease.

The median number of concomitant drugs was three (range 2–8); the majority included anti-hypertensive (beta blockers, Angiotensin-Converting Enzyme Inhibitors, calcium channels antagonists) and cholesterol-lowering drugs (statins).

An objective response was observed in 18 patients, with 3 patients showing a complete response, and 10 patients showing a partial response. Five patients had a stable disease for >6 months and two patients showed a progressive disease. No further therapies were prescribed in these patients. After a median follow-up of 36 months, the median duration of response was longer than 1 year.

No patients reported grade 3 or higher treatment-related adverse events. Grade 2 adverse events were reported in five patients (25%). Adverse events included 12% pneumonia, 12% diarrhea, and an increase in alanine transaminase and aspartate transaminase. Grade 1 toxicity was recorded in 50% of the population (diarrhea and fatigue were observed in four patients each, and myalgia was observed in two patients).

Overall, treatment was well tolerated, with 10% of patients discontinuing therapy not due to treatment-related adverse events (one hip fracture and one heart failure).

At the time of the present analysis, twelve patients (60%) are currently on treatment with maintained disease control. The median duration of treatment was 15 months. Figure 1, Figure 2, Figure 3 and Figure 4 report pre- and post-treatment imaging (stable disease, progressive disease, and objective partial and complete response achieved).

## 4. Discussion

The management of cSCCs requires a multidisciplinary approach. With the advent of Cemiplimab, the outcomes of patients with locally advanced/metastatic cSCC have significantly improved, and the indication for repeated local treatment (e.g., surgery, radiotherapy) against systemic therapy should always be discussed in a multidisciplinary setting.

Ultra-octogenarian patients with cSCC are potentially frail, and adequate selection for systemic treatment is mandatory. According to our experience, most patients with LAcSCC are elderly or frail with multiple comorbidities. Frailty is defined as an age-related condition of increased vulnerability to acute endogenous or exogenous stressors [8]. Therefore, we are used to discussing in the MDT an adequate approach of aging, polypharmacy, and comorbidities.

Biological age does not represent a contraindication for Cemiplimab, but a comprehensive assessment of frailty is required in this population. So, scientific societies recommend CGA for patients older than 70 years old. However, more than 50% of all solid tumors and more than 80% of cSCC occur in patients older than 70 years old, making the systematic implementation of CGA difficult to perform in daily clinical practice [4]. Recently, the International Society of Geriatric Oncology published the recommendations on skin cancer management in older patients. Based on a systematic literature review of 154 selected articles, Rembielak et al. concluded that patient age should not be the sole deciding factor and MDT assessment is crucial for all patients. It is advisable to offer a comprehensive geriatric assessment, and patients should be actively part of the MDT discussion [11].

Among screening tools, we used the G8 tool, which is a simple tool to routinely identify patients who should have a complete assessment in geriatric oncology [8]. To assess the percentage of frail patients, we tried to correlate the G8 score with the Frailty assessment tool (FRAIL scale) [10].

We found the G8 to be a very useful screening tool in the clinical practice to target a geriatric assessment, and it was more precise than the FRAIL score, which did not separate prognostic and comorbidity in our population. Indeed, most of our patients had 0 to 4 illness and weight loss as well as fatigue.

Cemiplimab has significantly changed the clinical armamentarium of locally advanced/metastatic cSCC, as safety and objective response rates justify treatment also in elderly and frail patients, who were historically not candidates for cytotoxic chemotherapy [12].

An adequate selection of patients before immunotherapy should be obtained in the context of an MDT. Some clinical predictors to define immunotherapy responses might be considered, such as the G8 score, obesity, polypharmacy, comorbidities, and a history of immunocompromised conditions [13].

Concerning efficacy, our study confirms that the ORR in this special population of patients is similar to that reported in other real-world studies.

As a retrospective case series, our data are in line with previous studies reporting data in the real-world.

Table 4 summarizes real-world data from the literature; in most studies, patients had a median age of 75 years (range: 71–83 years), with only a small percentage of ultra-octogenarians and special population.

A previous Italian multicenter experience reported data on 131 patients with a median age of 79 years (range 19–95) and a history of immune suppression in nearly 20% of patients (9.2% with concurrent chronic lymphoproliferative disease and 8.5% with concomitant autoimmune disease) [16]. In this case series, head and neck primary cSCC and hemoglobin values in the normal range were significantly associated with a better response to Cemiplimab. Regarding cSCCs on the genitalia, antibiotic therapy within 1 month from Cemiplimab initiation, ECOG PS ≥ 1, chronic corticosteroids therapy, previous radiation therapy to lymph nodes, and previous chemotherapy were significantly associated with a worse response [16]. Similarly, in our case series, patients with primary cSCC in the arm or the back (n = 6) had the worst outcome, as compared with the other primary site’s cSCC.

In the German ADOReg Registry, 9 out of 39 patients were receiving immune-suppressive therapy during treatment with Cemiplimab. No significant differences between immune-competent and immunosuppressed patients (48.1 vs. 50.0%, respectively) were registered, although these responses less often resulted in durable remissions [24]. In this study, the tumor response rate was 48.6% and the median PFS in the whole population was 29.0 months (PFS among immune-suppressed patients was 9 months) [24]. Our data confirm good response rates among immune-suppressed patients, although the small sample size and the short follow-up do not allow definite inference.

As in other solid tumors, complete response is associated with durable remission, likely sustained by significant changes in the tumor microenvironment [25]. In the German real-world experiences of six skin cancer centers, Salzmann et al. confirmed durable remissions among responding patients. The response was independent from the PD-1 inhibitor used (i.e., Cemiplimab vs. Nivolumab vs. Pembrolizumab) and the disease stage (i.e., locally advanced or metastatic). Among predictive factors for disease response, the primary tumor of the inferior limb and elevated serum lactate dehydrogenase levels at baseline correlated with poorer outcomes [15].

Due to the small number of patients, we cannot confirm a correlation between outcome and lactate dehydrogenase levels. According to other real-world experiences, the response rates among elderly patients with cSCC range from 32 to 80%. This might depend on patient characteristics, including age, disease stage, number of previous local treatments received, comorbidity, and immune- suppression. It must be stressed that in these published series, the range of age differs as well as the number of immunosuppressed patients and special population definition.

In the Israelian experience of 102 patients treated with PD-1 inhibitors (Cemiplimab or Pembrolizumab), the response rate was higher than other real-life experiences (80%), with a complete response in 45.2% and partial response in 35.5% of the patients. According to this data, the authors supported immune checkpoint inhibitors to be amenable for use in elderly or frail patients with comorbidities [23].

The phase IV CASE study is a big, non-interventional, survivorship and epidemiology study with Cemiplimab in a real-world setting which is currently ongoing. Patients’ enrolment is expected to be completed by September 2025 [26]. Preliminary data from this study were presented at the ESMO annual meeting in 2022: Considering the first timepoint (188 enrolled patients), the safety, tolerability, and effectiveness of Cemiplimab were consistent with the results observed in the registration clinical trial (i.e., ORR 42.1%; treatment-related adverse event incidence: 25.3%; incidence of G3–4 adverse events: 4.3%) [27].

Also, data from the French experience on 32 patients confirmed the other real-world experience (ORR 67%, with 33% complete response) [19]. In contrast, Valentin et al. reported a high discontinuation rate (41%) with 32% ORR, 47% DCR, and 35% PD in 23 older patients (median age of 83 years old) [20].

A higher disease control rate (about 50%), in line with the previous experiences, was reported by Ríos-Viñuela [22]. All these studies confirmed a strong and durable response among all patients with LAcSCC. However, in immunosuppressed patients, these responses less often resulted in durable remissions [22].

In a retrospective study on 465 cSCC patients, both immunocompetent and immune-suppressed, Zavdy et al. demonstrated lower survival rates, higher rates of positive resection margins, higher recurrence rates, and multiple cSCC tumors in the immune-suppressed patients. Aetiologies for immunosuppression in this study included transplantation, chronic lymphatic leukemia, chronic kidney disease, psoriasis, rheumatoid arthritis, and systemic lupus erythematosus. Transplant recipients had multiple cSCC tumors (35%), with the highest number of primary tumors compared to controls, but also compared to all other immunosuppressed groups. The number of lesions and the advanced stages might be explained by the number of surgical procedures proposed [12].

The strength of our series is that all patients were evaluated in an MDT setting specialized in skin cancer tumors, with long-standing experience, and the same systemic treatment was offered to all patients. The limitations of our study include its retrospective nature, the small sample size, and the length of follow-up. Further prospective data collection is warranted to collect more data on Cemiplimab treatment in this group of patients.

## 5. Conclusions

cSCC is extremely common among ultra-octogenarian patients. Immunotherapy with anti PD-1 monoclonal antibody Cemiplimab can provide significant clinical benefits for patients with LAcSCC or metastatic cSCC who are not candidates for curative surgery or radiation. Elderly patients with comorbidities, or treated with concomitant immune suppression, are excluded from clinical trials. However, they represent a significant proportion of patients in everyday clinical practice. Indeed, not all centers can routinely perform a geriatric assessment.

The G8 tool might help to identify elderly cancer patients who could benefit from CGA.

In conclusion, our data suggest that Cemiplimab is also feasible in frail patients (both for age and/or comorbidities); a multidisciplinary approach is strongly recommended to correctly address patients to systemic treatment, if indicated.

## Figures and Tables

**Figure 1 vaccines-11-01500-f001:**
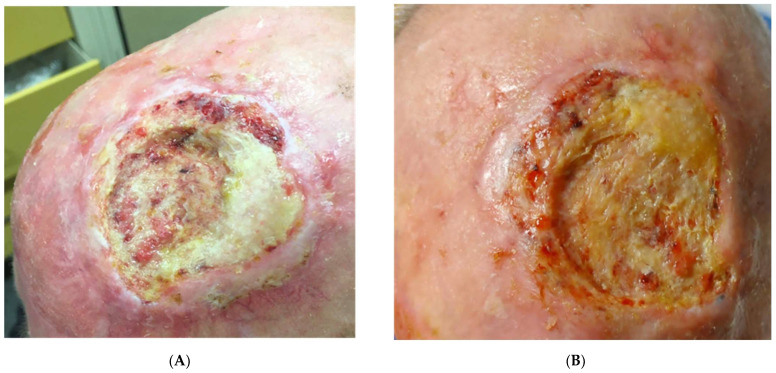
Patient 7. Before treatment, a 7-cm ulcerated nodule on the scalp, with infiltrated edges (**A**). After three cycles of Cemiplimab (**B**), the lesion was stable in its size, but slightly less infiltrated (Stable Disease).

**Figure 2 vaccines-11-01500-f002:**
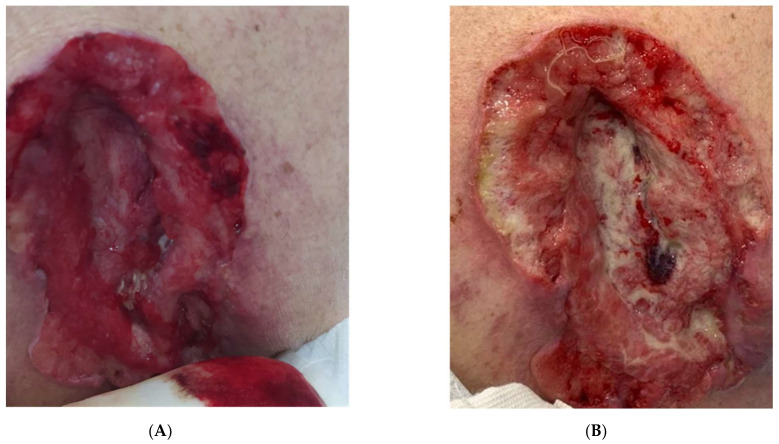
Patient 9. A wide ulcerated lesion involving subcutaneous and muscle tissue, located in the lumbar region (**A**). After five cycles of Cemiplimab (**B**), the persistence of the ulcer can be appreciated, but deeper and moister. (Progressive disease).

**Figure 3 vaccines-11-01500-f003:**
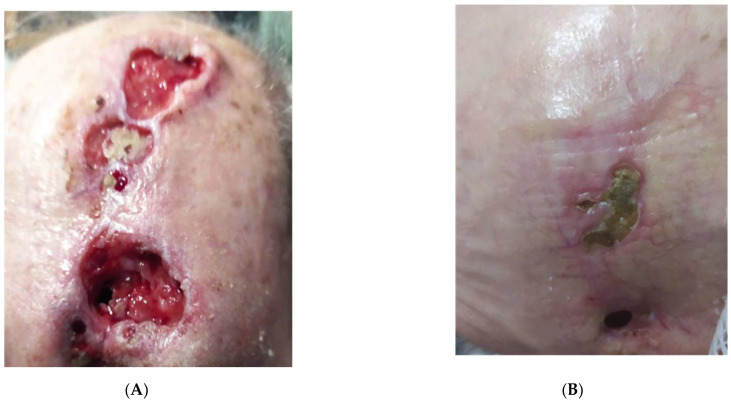
Patient 12. Three ulcerated nodules located on the scalp, of various sizes, one of which involves the cranial bone (**A**). After four cycles of Cemiplimab (**B**), two ulcers healed and were covered with yellowish crusts, while the third did not show signs of persistency at dermoscopy. (Partial response).

**Figure 4 vaccines-11-01500-f004:**
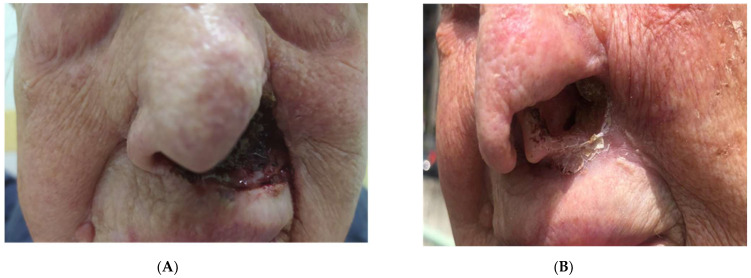
Patient 19. At the left nasolabial fold, a wide erythematous nodule, 5 cm in diameter, that destroyed the nasal cartilage (**A**). After four cycles of Cemiplimab (**B**), the nodule healed with a scar, while on the nasal fold, the loss of substance was visible in the absence of persistence of disease. (Complete response).

**Table 1 vaccines-11-01500-t001:** G8 score and FRAIL scale [9,10].

G8	FRAIL
Food intake over the past 3 months due to loss of appetite, digestive problems, chewing or swallowing difficulties (range no decrease = +2 severe decrease = 0)	Fatigue A = 1 = most of the time; B = 0 = a little
Weight loss during the last 3 months (range >3 kg = 0; no loss = +3)	Resistance: difficulty walking up 10 steps Y = 1 N = 0
Mobility (range bed or chair = 0; goes out = +2)	Ambulation: difficulty walking 300 m or a block y = 1 N = 0
Neuropsychological conditions (severe dementia = 0 no limits = + 2)	Illness * (0–4 = 0; 5–11 = 1)
Body mass index (<19 kg/m^2^ = 0 ≥ 23 kg/m^2^ = +3)	Loss of weight (y = 1 No = 0)
Polypharmacy (≥3) no = +1; yes = 0	
How do you consider the health status (not as good = 0; better = 2)	
Age >85 = 0; 80–85 = 1 < 80 = 2	
Range 17 low risk ≥ 14	Range 5 High risk

* including hypertension, diabetes, cancer, chronic lung disease, heart attack, congestive heart, angina, asthma, arthritis, kidney disease.

**Table 2 vaccines-11-01500-t002:** Baseline clinical characteristics of the study population (N° = 20).

Patients	Site	Age	Histology	Grading	Comorbidity *	Pharmacologic Therapy **	G8 Score	FRAIL Score	ECOG PS	Imm- Suppr
1	hnc	90	SCC	2	3	1	12	2	2	
2	hnc	85	SCC	3	1	1	9	3	1	
3	hnc	91	SCC	2	1,2,3	2	15	2	2	
4	back	86	SCC	2	1,2,3	2	14	2	2	
5	hnc	87	SCC	2	1	2	9	4	1	
6	back, hnc, arm	88	BCC + SCC	2	1,2,3	2	12	2	3	Y ^1^
7	hnc	84	SCC	3	1,2,3	3	14	2	2	Y ^2^
8	hnc	85	SCC	2	3	2	14	2	1	
9	arm	88	SCC	3	1	1	12	3	2	
10	hnc	80	SCC	2	1	1	10	4	1	
11	hnc	82	SCC	2	0	0	14	3	1	
12	hnc	81	SCC	2	3	2	15	2	1	
13	hnc	80	SCC	2	3	2	10	3	2	
14	hnc	86	SCC	3	0	2	14	2	1	
15	arm	87	SCC	2	1,2,4	1	9	4	2	
16	arm	81	SCC	2	3	2	9	3	3	
17	hnc	86	BCC + SCC	2	1,3	2	14	2	1	
18	hnc	98	SCC	2	1	3	9	2	3	
19	hnc	103	SCC	3	0	0	14	2	1	
20	arm	90	BCC + SCC	2	1	2	12	2	2	

Abbreviations: HNC head and neck cancer; SCC squamous cell carcinoma; BCC basal cell carcinoma. * Comorbidity: 1 cardiovascular; 2 respiratory; 3 metabolic; 4 neurologic. ** Pharmacologic therapy: 1 = 1–3 drugs; 2 = 4–6 drugs; 3 > 6 drugs. ^1^ Myelofibromatosis ^2^ Chronic lymphocytic leukemia.

**Table 3 vaccines-11-01500-t003:** Overview of treatment characteristics and safety in the study population.

Pts	N° of Surgery	RT	RT Doses Gy	Disease Stage	Cycles N°	Tox G1	Tox G2	Response
1	2	N	-	III	6	Diarrhea	Pneumonitis	PD
2	3	Y	50	IV M0	27			CR
3	2	Y	60	III	12	Myalgia	Diarrhea	PR
4	2	Y	56	IV M0	7	Diarrhea		PR
5	3	Y	54	IV M1	12	Fatigue	Transaminitis	SD
6	3	Y	54	III	7	Diarrhea	Pneumonitis	SD
7	3	Y	60	IV M0	7			SD
8	4	Y	66	III	8	Myalgia		PR
9	4	Y	54	IV M1	10			PD
10	3	Y	56	IV M0	30			PR
11	3	N		IV M0	6	Fatigue		SD
12	3	Y	60	IV M0	9			PR
13	3	Y	60	IV M0	5			PR
14	1	N		IV M0	13	Fatigue		PR
15	1	N		IV M0	3		Diarrhea	PR
16	2	N		IV M0	6			CR
17	3	N		IV M0	7	Fatigue		CR
18	1	Y	66	IV M0	2	Diarrhea		CR
19	3	Y	60	IV M0	6	Creatinine increase		CR
20	4	N		IV M1	12			PR

Abbreviations: CR, complete response; PR, partial response; PD, progressive disease; SD, stable disease Tox G1 = toxicity grade 1; Tox G2 = toxicity grade 2.

**Table 4 vaccines-11-01500-t004:** Real-world data of Cemiplimab in cSCC (source: Pubmed; accessed on 22 July 2023).

Study	Type	Patients (n.)	Median Age, Years (Range)	Response	Toxicity, Any Grade (%)	Special Population
Rischin D 2020 [14]	Phase II	114	71 (38–90)	ORR 51%	22 G3–4 27 G2	NR
Salzmann MM 2020 [15]	Retrospective observational multicenter	46	76 (39–92)	ORR 58.7% DCR 80.4%	13 G3–4 8.7 discontinuation	
Baggi A 2021 [16]	Retrospective observational multicenter	131	79 (19–95)	ORR 58% DCR 79%	9.2 G3–4	17.7% *
Strippoli S 2021 [17]	Retrospective observational monocentric	30	81 (36–95)	ORR 76.7% CR 30%	10 G3–4 33 G2	16%
Hober C 2021 [18]	Retrospective observational multicenter	245	77 (64–90)	ORR 48.6% 1 y OS 73% vs. 36%, for pts with PS < 2 vs. ≥ 2	9 G3–4 31 G1–2	21%
Guillaume T 2021 [19]	Retrospective observational monocentric	18	80 (45–96)	ORR 67% CR 33%	8 G3–4 33 G2	16.7%
Valentin J 2021 [20]	Retrospective observational monocentric	22	83 (55–93)	ORR 32% DCR 79%	45 G3–4 32 G1–2 41 discontinuations	36%
Bailly Caillé 2023 [21]	Retrospective observational monocentric	33 (12 + RT)	75 (63–88)	ORR 45.5% DCR 70%	23 G3–4 29 G1–2	NR
Ríos-Viñuela E 2023 [22]	Retrospective observational monocentric	13	81 (56–91)	ORR 62%	0 G3–4 46 G1–2	NR
Averbuch I 2023 [23]	Retrospective observational	102	78.5 (51–96)	mPFS 29.5 m ORR 80.6% CR 45.2%	5 G3–4 55 G1–2	

* 9.2% had a concurrent chronic lymphoproliferative disease and 8.5% had a concomitant autoimmune disease. Abbreviations: CR, complete response; DCR, disease control rate; ORR, overall response rate; PFS, progression free survival; G, grade.

## Data Availability

Data are available at the Hospital archive.

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
