# Peer review of "Cemiplimab in Ultra-Octogenarian Patients with Cutaneous Squamous Cell Carcinoma: The Real-Life Experience of a Tertiary Referral Center"

_vaccines, 2023, doi:10.3390/vaccines11091500_

Round 1

Reviewer 1 Report

The present study conducted by Denaro et al, showed the outcomes of Cemiplimab in elderly patients with locally advanced/metastatic LAcSCC/mcSCC treated in their tertiary referral center. The author assessed the real-world effectiveness of this drug and summarized the findings using appropriate assessment tools. However, I have some concerns about this study:

Comments:

1. The abstract lacks proper structure. The results section of the abstract should be rewritten to enhance clarity.

2. The manuscript's structure needs improvement. The exclusion and inclusion criteria are separately mentioned in both the introduction and methods sections. These criteria should be placed under the Method section.

3. The term "FRAIL" should be expanded upon by the author, preferably at the beginning of the text where they mentioned it first.

4. The context of Table 2 is unclear; it's uncertain whether it represents literature data or data from the current study. If it pertains to the current study, this should be explicitly mentioned in the results section.

5. The author should meticulously review the manuscript for grammatical accuracy and coherence. Some sentences lack clarity and fail to convey a meaningful message.

The author should meticulously review the manuscript for grammatical accuracy and coherence. Some sentences lack clarity and fail to convey a meaningful message.

Author Response

Dear reviewer

Thank you for giving us the opportunity to submit a revised draft of the manuscript.

We appreciate the time and effort that you and the reviewers dedicated to providing feedback on our manuscript and are grateful for the insightful comments on and valuable improvements to our paper. We have incorporated most of the suggestions made by the reviewers. The changes are highlighted within the manuscript. Please see below

The present study conducted by Denaro et al, showed the outcomes of Cemiplimab in elderly patients with locally advanced/metastatic LAcSCC/mcSCC treated in their tertiary referral center. The author assessed the real-world effectiveness of this drug and summarized the findings using appropriate assessment tools. However, I have some concerns about this study:

Comment: The abstract lacks proper structure. The results section of the abstract should be rewritten to enhance clarity.

Response: In order to comply with the reviewer’s suggestion, we changed the abstract structure and rewrote the results section, in order to improve its readability and provide more clear data.

Lines 25-31

Comment: The manuscript's structure needs improvement. The exclusion and inclusion criteria are separately mentioned in both the introduction and methods sections. These criteria should be placed under the Method section.

Response: We revised the manuscript structure as suggested. In order to avoid misunderstanding, we specified the inclusion and exclusion criteria only in the methods part (Lines 159-163). The exclusion criteria mentioned in the Introduction section refer to the exclusion criteria of ENPOWER CSCC1 trial. Now, the section has been corrected as follows: Exclusion criteria of ENPOWER CSCC1.

Comment: The term "FRAIL" should be expanded upon by the author, preferably at the beginning of the text where they mentioned it first.

Response: As suggested by the reviewer, we expanded the term FRAIL (lines 121-122).

Comment: The context of Table 2 is unclear; it's uncertain whether it represents literature data or data from the current study. If it pertains to the current study, this should be explicitly mentioned in the results section.

Response: we thank the reviewer for this suggestion. Table 1 represents literature data, so we clarified this in the title.

Comment: The author should meticulously review the manuscript for grammatical accuracy and coherence. Some sentences lack clarity and fail to convey a meaningful message.

Response: in order to comply with the reviewer’s comment, we thoroughly revised the English form of the entire manuscript.

Reviewer 2 Report

In this original article, the authors review the real-world experience of treating with the anti-programmed death 1 (PD-1) antibody Cemiplimab 20 ultra-octogenarian patients with locally advanced/ metastatic cutaneous squamous cell carcinoma (LAcSCC/ msSCC) of sun-exposed areas.  Clinical benefit from this systemic therapy was achieved in most patients, tumor shrinkage and pain relief were reported, and severe adverse events were not observed.

Unfortunately, the manuscript is poorly organized and poorly written.  In particular, the article does not clearly demonstrate in which way it is an original contribution to the field, especially given a previous case series of 131 patients that included patients older than 80 (Ref. 10).  The description of the methods is incomplete. It is unclear if this study is indeed a retrospective study because informed consent was obtained from all study patients.  The results section does not describe the figures, and does not refer to Tables 3-4 and Figures 1-4 at the proper places in the text.  Although the results appear to support the main conclusion, they are not thoroughly discussed and compared with other published cases. The review of the literature at the beginning of the discussion should be moved to the end of that section along with Table 1.  The conclusion is unclear and unnecessarily long.

Line 54-65. Describe the mechanism of action, response rate and adverse effects of Cemiplimab in more detail with quantitative data and supporting references.

Line 77-79. Why were the Geriatric 8 scale and FRAIL scale chosen for this study?  Have they been validated in ultra-octogenarian patients?

Line 83. Renumber Table 2 as Table 1 and describe in the methods section.

Line 85-87, 102-103. If this study is a retrospective study, why was informed consent obtained from all study patients?  Typically, the requirement for informed consent is waived by institutional review board/ethics committee for retrospective studies.

Line 88-90. What were the patient exclusion criteria?

Line 97-101. Describe the Geriatric 8 scale and FRAIL scale in more detail.  Were they both administered only once prior to treatment start?  Also describe comprehensive geriatric assessment (CGA, line113) here.

Line 178-238. Discuss all pertinent publications about Cemiplimab in LAcSCC/ msSCC in chronological order.

Line 239-275. Discuss the methods, the results and how they compare to previous studies in that order.

Table 1. List the references in chronological order.  Add the age range.  Correct the spelling of “Bailly Caillé” and “Guillaume T”.  Be consistent with the use of a decimal period, instead of a comma, throughout the manuscript, e.g., “17.7%”. Describe the criteria for the toxicity grades in the methods section.

Table 2. Correct the units of body mass index “kg/m2”.  Define criteria of the illness scale 0-11 in the methods section.

Table 4. Define Tox G1 and Tox G2 as toxicity grade 1 and 2 in a table footnote.

Figure 1-4. Add a description of the images in figure legends in addition to the figure titles.  Indicate the size of the lesions in the legend or in the figure using a scale bar. Mention the duration of the lesion and the type and duration of treatment applied.  Figure 4A should be pre-treatment.

Line 305-309. What do you mean by “a communication was done”?  If this study is a retrospective study, why did the informed consent was obtained from all study patients?  Typically, the requirement for informed consent is waived by institutional review board/ethics committee for retrospective studies.  That is not what these two statements say.  What was actually done?

English language editing required for spelling and grammatical errors and incorrect word usage. The flow of the text also needs to be improved.

Author Response

Dear Editors and dear reviewers

Thank you for giving us the opportunity to submit a revised draft of the manuscript.

We appreciate the time and effort that you and the reviewers dedicated to providing feedback on our manuscript and are grateful for the insightful comments on and valuable improvements to our paper. We have incorporated most of the suggestions made by the reviewers. The changes are highlighted within the manuscript. Please see belowReviewer2

In this original article, the authors review the real-world experience of treating with the anti-programmed death 1 (PD-1) antibody Cemiplimab 20 ultra-octogenarian patients with locally advanced/ metastatic cutaneous squamous cell carcinoma (LAcSCC/ msSCC) of sun-exposed areas.  Clinical benefit from this systemic therapy was achieved in most patients, tumor shrinkage and pain relief were reported, and severe adverse events were not observed.

Comment: Unfortunately, the manuscript is poorly organized and poorly written. 

Response: We thank the reviewer for the comments: we revised and modified the manuscript structure, as suggested.

Comment: In particular, the article does not clearly demonstrate in which way it is an original contribution to the field, especially given a previous case series of 131 patients that included patients older than 80 (Ref. 10).  

Response: We agree with the reviewer’s observation, however our study considered only the ultra-octogenarians, while in the Italian retrospective observational multicenter study by Baggi A et al, the median age of the 131 patients was 79 years (range 19-95).

Comment: The description of the methods is incomplete. It is unclear if this study is indeed a retrospective study because informed consent was obtained from all study patients.

Response: This was a retrospective study. All patients provided written informed consent according to actual rules for data collecting processing and privacy required in Italy.

Comment: The results section does not describe the figures, and does not refer to Tables 3-4 and Figures 1-4 at the proper places in the text.  

Response: We reported tables 3-4 and Figures 1-4 in the results section (lines 113 and 149). Tables 3 and 4 summarize patients’ characteristics and treatments. Figures 1-4 report pre and post treatment imaging. We chose the most significant responses reported among patients enrolled in this study (i.e., stable disease, progressive disease and objective response achieved).

Comment: Although the results appear to support the main conclusion, they are not thoroughly discussed and compared with other published cases.

Response: in order to comply with the reviewer’s comment, we revised the discussion section accordingly.

Comment: The review of the literature at the beginning of the discussion should be moved to the end of that section along with Table 1.

Response: we thank the reviewer for this suggestion. The discussion was revised, and moved at the end the review of literature

 Comment: The conclusion is unclear and unnecessarily long.

Response: We revised the conclusion section and shortened it accordingly.

Comment: Line 54-65. Describe the mechanism of action, response rate and adverse effects of Cemiplimab in more detail with quantitative data and supporting references.

Response: we added a section reporting more information and data on the efficacy of cemiplimab in cuSCC (Lines 88-109).

Cemiplimab is Ig4 monoclonal antibody against PD1.It blocks the interaction among the receptor PD-1 and its ligands PD-L1 and PD-L2 removing the negative signal responsible for inhibition of T cell function such as proliferation, cytokine secretion, and cytotoxic activity. PD-1 acts as a brake that keeps T cells from creating an immune reaction against cSCC. Cemiplimab potentiates T cell responses, including anti-tumour response, through the blockade of PD-1 binding to PD-L1 and PD-L2 ligands. The landmark study of Cemiplimab for cSCC was ENPOWER CSCC1, in this study 193 patients were enrolled (Group 1, metastatic CSCC and Group 2, locally advanced CSCC received Cemiplimab 3 mg/kg IV every 2 weeks for up to 96 weeks; Group 3, metastatic CSCC, received Cemiplimab 350 mg IV every 3 weeks for up to 54 weeks). The study reached the primary endpoint with a benefit in objective response rate, ORR in total population 47,2% (ORR 50.8, 44.9,46.4% respectively for group1,2,3). At the final analysis of ESMO meeting 2022 OS at 48 months was 61.8% (95% CI: 54.0-68.7). Overall median DOR was 41.3 months. Fatigue (34.7%) was the most common treatment-emergent adverse event (TEAE) of any grade; hypertension (4.7%) was the most common Grade ≥3 TEAE. Median progression free survival was 22.1 months.[7] This study did not include special population. Exclusion criteria of ENPOWER CSCC1 included: autoimmune disease requiring systemic immunosuppressant agents within 5 years; history of solid organ transplant; history of pneumonitis within the last 5 years; active infection requiring therapy, including known infection with human immunodeficiency virus, or active infection with hepatitis B or hepatitis C virus; chronic lymphocytic leukaemia (CLL); brain metastases or Eastern Cooperative Oncology Group (ECOG) performance score (PS) ≥ 2.[6]

Comment: Line 77-79. Why were the Geriatric 8 scale and FRAIL scale chosen for this study?  Have they been validated in c?

Response: The G-8 screening tool and the FRAIL were developed to identify elderly cancer patients who would benefit from comprehensive geriatric assessment (CGA). The G-8 showed good screening properties for identifying elderly patients who could benefit from CGA. To the best of our knowledge, no specific validation in ultra-octogenarian are published. We specified this in the manuscript (Introduction section, Lines 114-126).

Comment: Line 83. Renumber Table 2 as Table 1 and describe in the methods section.

Response: We renumbered all the tables, and described them in the Methods section, as suggested.

Comment: Line 85-87, 102-103. If this study is a retrospective study, why was informed consent obtained from all study patients?  Typically, the requirement for informed consent is waived by institutional review board/ethics committee for retrospective studies.

Response: See previous comment : This is a real retrospective study. All patients provided written informed consent according to actual rules for data collecting processing and privacy required in Italy.

Comment: Line 88-90. What were the patient exclusion criteria?

Response: We specified the Exclusion Criteria in the Methods section (Lines 474-475).

Comment: Line 97-101. Describe the Geriatric 8 scale and FRAIL scale in more detail.  Were they both administered only once prior to treatment start?  Also describe comprehensive geriatric assessment (CGA, line113) here.

Response: we thank the reviewer for this important observation. More details regarding the G8 score and FRAIL score were added to the manuscript in the Introduction section (lines 110-126) and summarized in Table 1. Information regarding timing of geriatric assessment in the present study were provided (Lines 480-482).

Comment: Line 178-238. Discuss all pertinent publications about Cemiplimab in LAcSCC/ msSCC in chronological order.

Response: in order to comply with the reviewer’s suggestion, we discussed the publications in a chronological order.

Comment: Line 239-275. Discuss the methods, the results and how they compare to previous studies in that order.

Response: as suggested by the reviewer, we discussed and compared the previous studies with our experience.

Comment: Table 1. List the references in chronological order.  Add the age range.  Correct the spelling of “Bailly Caillé” and “Guillaume T”.  Be consistent with the use of a decimal period, instead of a comma, throughout the manuscript, e.g., “17.7%”. Describe the criteria for the toxicity grades in the methods section.

Response: in order to comply with the reviewer’s suggestion, we made all the suggested changes. We described the criteria for toxicity grades in the Methods section (Lines 483-485).

Comment: Table 2. Correct the units of body mass index “kg/m2”.  Define criteria of the illness scale 0-11 in the methods section.

Response: we thank the reviewer for this suggestion, we insert the number of illness in the methods section , and correct the body mass index units.

Comment: Table 4. Define Tox G1 and Tox G2 as toxicity grade 1 and 2 in a table footnote.

Response: we specified the definition in the table’s footnotes.

Round 2

Reviewer 2 Report

The revised manuscript shows substantial improvements.  Nevertheless, the following areas of concern must be addressed.

1. The figure legends are totally inadequate.  Describe briefly the clinical and treatment history of each patient as well as the size and depth of the main lesion(s) and their response to treatment in the figure legends.

2. The text of the discussion is still difficult to follow.  Please begin by discussing the methods (CGA, G8 tool, FRAIL tool...) and the experimental design of your study and comparing those to the literature.  Summarize your main findings and discuss them in the context of the literature.  Summarize in the text the main findings (Refs. 11-20) listed in Table 4.

3. Shorten the length of the conclusion by half.

There are still several English language spelling and grammatical errors. The flow of the text such as transitioning from one paragraph to the next also needs to be improved, 

Author Response

dear reviewer 

we revised the manuscript according to the suggestions.

Thank you

best regards

Nerina Denaro
